# The Brain in Context: A Scoping Review and Concept Definition of Neuro-Informed Policy and Practice

**DOI:** 10.3390/brainsci14121243

**Published:** 2024-12-11

**Authors:** Sally Staton, Laetitia Coles, George Normore, Charlotte Casey, Bonnie Searle, Sandy Houen, Azhar Potia, Rebecca Crompton, Deborah Long, Michael Hogan, Karen Thorpe

**Affiliations:** 1Queensland Brain Institute, The University of Queensland, St. Lucia, Brisbane 4072, Australia; l.coles@uq.edu.au (L.C.); g.normore@uq.edu.au (G.N.); charlotte.casey@uq.edu.au (C.C.); b.searle@uq.edu.au (B.S.); s.houen@uq.edu.au (S.H.); a.potia@uq.edu.au (A.P.); b.crompton@uq.edu.au (R.C.); k.thorpe@uq.edu.au (K.T.); 2School of Nursing, Queensland University of Technology, Brisbane 4059, Australia; da.long@qut.edu.au; 3Thriving Queensland Kids Partnership, Australian Research Alliance for Children and Youth, Brisbane 4001, Australia; michael.hogan@aracy.org.au

**Keywords:** neuroscience, policy, practice, brain development, review

## Abstract

Background/objectives: Among the developmental sciences, discovery in neuroscience has underpinned research innovations and made a significant contribution to knowledge translation. With the growth of neuroscience discovery, policymakers and practitioner workforces have adopted ‘neuro-informed’ in decisions targeting the delivery of human, social, and economic wellbeing. Methods: In this scoping review, we examined the use and conceptualization of neuro-informed policy and practice (NPP) over the last two decades. We aim to establish a working definition of NPP and identify the key knowledge bases underpinning the application of NPP, with a specific focus on children and young people. Results: A total of 116 publications related to NPP were identified across academic and policy sources. Publications derived from diverse fields (e.g., psychology, social policy, medicine, urban planning). Health and Education were the most common target areas for NPP; however, applications of NPP to social services, law, and physical environments were also identified. Despite the growth in NPP, concept definitions of NPP were limited and primarily tautological. A four-stage process of concept definition was used to develop a working definition of NPP applicable to different systems, workforces, and contexts. By applying content analysis, 12 distinct knowledge bases underpinning NPP were identified. Conclusion: Our scoping review highlights the importance of defining the concept of neuro-informed policy and practice, extending beyond the brain or individual in isolation to include consideration of the brain in context.

## 1. Introduction

The ground-breaking work From Neurons to Neighborhoods, published in 2000, drew new attention to the way in which understanding the biological and physiological underpinnings of brain development could inform not only our understanding of human health and functioning, but also translate this understanding into social, community, and economic actions [1]. For example, evidence that the quality of experiences in the first five years of life is the foundation of neural architecture (neuroscience) [2] that entrains ongoing trajectories of learning and development (developmental science) [3] has underpinned substantial investment into Early Education and Care programs globally [4,5,6]. Commensurate with such growth in the application of neuroscience has been the emergence of new terminologies, such as brain capital, brain health, the science of learning, and neuro-informed policy and practice (NPP). The first three of these terms have been described and defined elsewhere [7,8,9]. To date, despite growing use, the definition and conceptualization of neuro-informed policy and practice remains elusive.

How we define concepts matters. Clearly defined concepts underpin rigorous scholarship and are necessary to progress theory, research, and translation into policy and practice. Yet, the problem of poorly defined concepts in academic scholarship is one that is both long-standing and extensive [10]. Clear conceptual definitions, at their most basic level, answer the question “what are we talking about?” [11]; they encompass a clear description of the core properties or attributes of a concept and allow us to distinguish different concepts from one another. Lack of clarity in conceptual definition presents a range of risks that hinder progress within research and research translation. This is particularly the case when a field proliferates in the absence of clarity and precision of definition. Burman and colleagues [12] provide an example of the use of the term self-regulation. The authors identified no less than 447 associated terms, 88 closely related concepts, and 6 broad conceptual clusters to define this widely used and deceptively ‘simple’ concept. Such dispersed and imprecise concepts present risks, among these inappropriate or inconsistent operationalization in measurement, misattribution of the associations between measures and outcomes, unjust critique, or disregard or discrediting of a theoretically important construct [10].

Clear concept definitions matter beyond the academy. Such definitions are vital to the way in which concepts are understood, applied, and accepted within the broader community. When concepts are poorly defined, there is an increased risk of misinterpretation, misapplication, wholesale rejection, reification, or the perpetuation of myths [10,13]. When translated into policy or practice action, poorly defined concepts are at best unhelpful and at worst can lead to the misdirection of resources or indeed harm. Internationally, there has been growing interest in the ways in which different services and systems should work collectively to support human functioning and wellbeing [14,15]. Within this movement, there have been calls for the identification of shared purpose, knowledge, and language that can be applied across different fields and disciplines to support collective understandings and integration [16]. The concept of NPP has been proposed as a potential unifying frame that could be used to support such systems-level work [17]. In this scoping review, we aimed to (1) establish a working definition of NPP that can be applied to different systems, workforces, and contexts and (2) identify the key knowledge bases underpinning the application of NPP. Given the origins and salience of NPP within developmental sciences, a central focus of our work is NPP actions directed at children and young people.

## 2. Methods

### 2.1. Transparency and Openness

This scoping review followed the Preferred Reporting Items for Systematic Reviews and Meta-Analyses extension for the Scoping Reviews (PRISMA-ScR) protocol checklist [18]. A scoping review protocol was developed and prospectively registered on the Open Science Framework (https://doi.org/10.17605/OSF.IO/A4VCD) prior to review.

### 2.2. Scoping Review Research Questions

The scoping review focused on addressing three key research questions:How is NPP defined in the current academic literature and policy and practice documents?Are there unifying themes (attributes) across different definitions of NPP relevant to different systems, workforces, or contexts?What are the key components (knowledge bases) of NPP, with a particular focus on application to children and young people?

### 2.3. Search Strategy

An initial search of several key policy and practice documents [19,20,21,22,23] was undertaken by three authors to assist in the identification of related terms. From this initial search, a range of search terms that related to NPP were identified and used to inform the following final search string: (“neuro* inform*” OR “neuro* integrat*” OR “neuroscience inform*” OR “neuroscience inegrat*” OR “brain science inform*” OR “brain science integrat*” OR “biopsycho* inform*” OR “biopsycho* integrat*” OR “brain architecture” OR “develop* child*”) AND (policy OR practice OR program).

### 2.4. Selection of Studies

The PRISMA flow diagram in Figure 1 outlines the study selection process. Database searches of the peer-reviewed academic and grey literature were performed between 21st and 23rd March 2023, using the following databases: Scopus, Web of Science, APA PsycInfo via Ebscohost, Worldwidescience.org, Campbell Collaboration Online Library, ERIC, and WHO.

All studies published in English on or after the year 2000 (i.e., following the publication of the groundbreaking Neurons to Neighborhoods [1]) that contained (1) a definition of NPP, (2) a specific framework or approach to NPP, or (3) discussion and/or an example of NPP were included. Publication types included, but were not limited to, peer-reviewed journal articles, government and/or industry reports, policy documents, working/white papers, dissertations/theses, book chapters, and/or full-text conference papers. Whole books and publications that focused specifically on the application of neuro-informed business models (e.g., application to marketing, military policy and practice, and business leadership if not related to workforces that work with children and young people) were excluded.

Database extraction and screening were undertaken by three authors using Covidence (www.covidence.org; accessed on 8 December 2024), a web-based collaboration software platform that streamlines the production of systematic and other reviews. One author reviewed the title and abstract of each document for relevance against the inclusion and exclusion criteria. Two authors (GN and CC) then independently screened the full text of each document to determine eligibility against the pre-determined inclusion/exclusion criteria. Where required, conflicts were resolved by a third researcher (BS) using a consensus method. Outcomes and reasons for exclusion were documented.

### 2.5. Data Extraction

Data were extracted from each document and comprised (i) publication characteristics, including title, author/s, publication date, type, and affiliation, (ii) publication field (e.g., psychology, education, health), (iii) terms used to denote NPP, (iv) included definitions of NPP, (v) key elements of NPP described (e.g., frameworks, content), and (vi) direct references of NPP as it relates to children or young people.

### 2.6. Data Analysis

Data were synthesized using quantitative (descriptive) and qualitative (thematic, content) approaches. To provide a framework for understanding the current application of NPP across fields and disciplines, the focus areas for each publication were examined and coded into super-ordinate target areas based on the broad fields in which NPP is applied (i.e., health, education, social services, law, and/or physical environments). The nature of the application (i.e., policy and/or practice) and whether the focus was on critiquing NPP was also coded. The key components of NPP were identified through a process of content analysis in which the key elements described in each publication were first identified and then grouped into common themes.

### 2.7. Development of Concept Definitions

A concept definition was generated by applying the four-stage methodology for creating concept definitions described by Podsakoff et al. [10]. In the first stage, all available definitions of NPP extracted from included publications were collated and key attributes used within each definition were identified. In the second stage, attributes were organized by themes and examined to determine which were necessary and sufficient attributes for defining NPP. This process was undertaken through comparison with three related terms and definitions selected based on their theoretical proximity and alignment to NPP. The three concepts and related definitions used for attribution comparison and distinction are provided in Table 1. Third, we developed a preliminary definition of NPP based on the attributes identified as necessary and sufficient. In our final stage, we refined our conceptual definition through consultation with experts across several relevant fields (e.g., Neuroscience, Psychology, Education, Sociology, Health, Community Development, Pediatrics).

## 3. Results

### 3.1. Search, Article Selection, and Extracted Data

The process of data screening is described in Figure 1. Following the removal of duplicates, database searches identified a total of N = 3697 records for screening, of which N = 329 publications underwent full-text review. A total of N = 116 publications met the inclusion and exclusion criteria following this process and were included in the data synthesis (see Appendix A for details of all included publications).

### 3.2. Publication Characteristics

While our search was conducted for publications between the year 2000 and March 2023, over 60% (N = 74) of included publications emerged from 2015 onward (Figure 2), with a noticeable increase in publications focused on NPP application to Health after 2015. Most publications were authored or published in the USA (N = 78, 67%), with Australia (N = 12, 10%) and the United Kingdom (N = 10, 9%) having the second and third highest rates of publication, respectively. The remaining studies were from the regions of South America, Asia, Europe, and Scandinavia. Publication types were primarily peer-reviewed journal articles (N = 71; 61%), book chapters (N = 24; 21%), and government or industry reports (N = 11; 9%). Publications of NPP emerged from a range of fields, including clinical psychology, counseling, developmental psychology, social policy, education, public health, nursing, medicine, nutrition, law, psychiatry, psychotherapy, social work, art therapy, educational psychology, economics, urban planning, religion, and science communication. A total of N = 77 (66%) publications made specific reference to children, young people, and/or workforces relevant to these groups.

#### 3.2.1. Target Areas

Five super-ordinate target areas for the application of NPP emerged. These were the areas of Health, Education, Social Services, Law, and Physical Environments. A small number of publications (N = 19, 16%) incorporated application to more than one target area (e.g., Health and Education) or across both policy and practice (N = 28, 24%); thus, the numbers presented below are not mutually exclusive.

#### 3.2.2. Health

Health was the largest target area identified across publications (N = 63, 54%) and primarily focused on health practice (N = 61, 97%) and, to a lesser extent, health policy (N = 16, 25%). Applications in health were largely related to the use or evaluation of models or treatments that incorporated neuroscience principles, particularly in the context of mental health conditions (e.g., Post-Traumatic Stress Disorder, Anti-Social Personality Disorder, Schizophrenia, Substance Use Disorders). For example, publications were focused on the application and evaluation of cognitive behavioral therapy (n-CBT) [28,29,30,31], psychotherapy [32], mindfulness [33,34], compassion-focused therapy [35], prolonged exposure therapy [36,37,38], music therapy [39,40], and art therapy [30,41]. Five publications discussed the use of the Neurosequential Model of Therapeutics [42,43,44,45,46]. Publications also included those focused on the application of evidence from neuroscience in health promotion, e.g., [47], early intervention, e.g., [48], perinatal and infant health, e.g., [49], nutrition, e.g., [50], and the diagnosis of specific health conditions, e.g., [51].

#### 3.2.3. Education

Education was identified as a target area in 51 publications (44%). Almost all included a focus on education practice (N = 48, 94%). Half of these publications also focused on education policy (N = 24; 47%). NPP was applied across a range of education contexts and student age groups, including schools, early childhood education and care settings, universities and tertiary training programs, and professional development delivery. The range of publications included those focused on the application and evaluation of neuroscience-informed programs and interventions to promote learning, e.g., [39,52,53], social-emotional functioning, e.g., [54,55,56,57], and learner engagement, e.g., [58]. Publications also discussed how understanding neuroscience can inform teaching practices and learning environments for optimal learning outcomes, e.g., [59,60], ensure reasonable expectations for children and young people at different ages, e.g., [61], and provide context for understanding how experiences, such as toxic stress, can influence learning and behavior, e.g., [62]. A total of 15 (29%) publications had a specific focus on critiquing the application of NPP in education contexts, which are discussed further in the critical considerations Section 3.3 below.

#### 3.2.4. Social Services

Twenty-eight (24%) publications were identified as having a Social Services focus. Within this category, there was a slightly greater focus on social service policy (N = 24, 86%) than on practice (N = 19, 68%). Key examples of neuro-informed policies discussed within these publications included those focused on (1) optimizing caregiving environments and conditions [48,63,64,65,66,67,68,69,70], (2) supporting parents and families via formal and informal social supports [68,71], (3) encouraging community-focused and early intervention strategies [71,72,73,74], (4) reducing maltreatment and informing child protection decisions [20,68,73,75,76], and (5) reducing poverty and economic inequality [68,72].

#### 3.2.5. Law and Built Environments

A small number of publications were identified within the target areas of law (N = 4) and built environments (N = 2). Three publications [77,78,79] within the area of law focused on how neuroscience can be applied to practice, for example, in determining criminal culpability or consideration of whether someone is classified as an adult, dangerous, cognitively impaired, or drug addicted. Two publications [73,80] that focused on the built environment looked at the application of neuroscience within the context of urban planning.

### 3.3. Critical Considerations for the Application of NPP

Twenty-four publications (21%) identified critical considerations regarding the ways in which neuroscience is understood and applied within the target areas of Education (N = 15), Health (N = 5), Social Services (N = 3), and Law (N = 2). Broadly, these critiques reflected the diversity of interpretations of NPP, particularly when these are focused on the brain in isolation or the individual in isolation, without considering the ways in which broader environments impact brain development.

#### 3.3.1. The Brain in Isolation

The first group of critiques emerged in response to the application and conceptualization of NPP as primarily focused on the brain in isolation. These critiques were particularly evident in publications targeted to Education, where NPP was conceptualized as the uncritical adoption of brain-based practices, such as brain-training or brain-centered learning strategies [81], with the absence of consideration of other aspects of learning and behavior. The key criticism of NPP in this context was that such approaches often only address a single aspect of learning (e.g., a focus on developing ‘intelligence’) [82], yet disregard the complexities of teaching and learning processes and relationships [54]. Narrow conceptualizations and applications of NPP and their potential to reinforce or perpetuate neuro-myths [55,61,65] also raised a range of ethical concerns. Concerns included those related to (1) grouping individuals into fixed neurological/brain-based categories (e.g., ‘brain-typing’) [81], (2) ignoring alternate forms of knowledge, agency, and diversity that can lead to the exclusion of some learners [82], (3) inequality in accessing science-based interventions, and (4) the privileging of neuro-typical brains [83,84,85]. Critiques were also evident beyond the context of education, in, for example, the application of NPP within the criminal justice system. These critiques specifically highlighted risks associated with decisions that are based upon incorrect understandings of neuroscience or an over-reliance on biological bases for criminal behavior, including the potential for the misuse of brain imaging techniques to persuade a judge or jury of innocence or guilt [77,79].

#### 3.3.2. The Individual in Isolation

The second group of critiques emerged in response to the application and conceptualization of NPP as primarily focused on the individual in isolation. These critiques were particularly evident in publications targeted to Health and Social Services where NPP was conceptualized as providing explanations for individual actions and behaviors, without consideration of broader social and relational contexts. The key criticism of NPP in this context was that such approaches oversimplify complex neuroscientific concepts and over-emphasize the importance of personal responsibility for individual behaviors (e.g., in addiction treatment) [47,86,87]. In the absence of consideration of the social and environmental context, NPP is criticized as preferencing a focus on behavioral interventions targeting individuals and disregarding the value of system, family, and community level supports [88].

### 3.4. Definitions of Neuro-Informed Policy and Practice

Across the identified papers, a range of different terminologies related to NPP were identified. Despite the diversity of terminologies used, specific definitions of key terms were limited. These definitions could be broadly grouped into three types: (1) those that provided general statement definitions; (2) those that provided a field-specific definition; and (3) those that provided more comprehensive definitions. At its simplest, general statement definitions—typically tautological in nature—defined NPP as “incorporates findings from neuroscience” [61] (p. 414), “the integration of neuroscience into …” [89] (p. 204), [90] (p. 93), [91] (p. 1), “incorporate[ing] neurobiological understandings” [28] (p. 211), “based on neuro-scientific discoveries” [88] (p. 198), or “based on neuroscience principles and an understanding of how the brain [works]” [83] (p. 16). Field-specific definitions were often accompanied by field-specific terminologies, such as neuro-counseling [51,91], neuro-education [58,83,92,93], and neuro-pedagogy [94]. Several papers cited Beeson and Field’s [51] (p. 74) definition of neuro-counseling, i.e., “the art and science of integrating neuroscience principles related to the nervous system and physiological processes underlying all human functioning…” e.g., [89] (p. 204). Neuro-education was defined as “translate[ing] relevant research findings from the neuro- and cognitive sciences [to] help educators interpret and apply these findings in the classroom” [95] (p. 137) and neuro-pedagogy as “when science and education meet, aiming for stimulating the brains of all types of learners” [94] (p. 27). A small number of papers provided more comprehensive definitions of NPP that, whilst derived within a particular health or education context, arguably lend themselves to application more broadly across fields. For example, King et al. [41] (p. 150) defined neuroscience informed as “… understanding of brain processes and functions related to human behavior and cognitive and emotional processing [that] can be incorporated into a … knowledge base that is used to underpin messages and metaphors”.

#### 3.4.1. Key Attributes of NPP

Across the available definitions of NPP, a total of 11 key attributes were identified as present within definitions of NPP (Table 2). Mapping of each attribute to NPP and related concepts (i.e., trauma-informed, evidence-based, and neuroscience) identified nine attributes (1, 2, 3, 5, 6, 7, 8, 10, and 11) deemed necessary (i.e., an essential property that all exemplars of the concept must possess) but not individually sufficient on their own to define the concept of NPP. The remaining two attributes (4—describes brain-body-environment interaction and 9—contains neuroscience messages and metaphors) were deemed as being sufficient (i.e., a unique property that only exemplars of the concept possess) but not necessary for defining the concept more broadly [10].

#### 3.4.2. NPP Concept Definition

The nine attributes deemed necessary and jointly sufficient for the conceptualization of NPP were used to generate a working definition of NPP that was revised through consultation. Given the identification of “brain health” as a key attribute essential to the concept definition of NPP, the WHO definition of brain health was incorporated within our definition to ensure clarification of this important component.

NPP is defined as the method and outcome of translating and applying current evidence from neuroscience and related fields about the processes underpinning human development and behavior to guide policy and practice actions. The intent of neuro-informed policy and practice is to create and promote optimal conditions for brain health* and (or) related positive physical, social, and community outcomes.

Where brain health is defined as: “the state of brain functioning across cognitive, sensory, social-emotional, behavioral, and motor domains, allowing a person to realize their full potential over the life course, irrespective of the presence or absence of disorders” [9] (Brain health).

### 3.5. Knowledge Basis of Neuro-Informed Policy and Practice

Content analysis of each of the publications, with a particular focus on those that included references to children and young people (N = 77) was undertaken. The analysis identified 12 key knowledge bases underpinning the application of NPP (summarized in Figure 3 and provided in more detail in Appendix A). Whilst the analysis for this section focused on NPP in relation to children and young people, analysis of all publications (N = 116), including those without specific reference to children and young people, identified similar knowledge bases, with the only distinguishing factor being a greater emphasis on drug use and addiction, e.g., [37].

## 4. Discussion

The publication From Neurons to Neighborhoods made clear that human development and psychological functioning occur in context and have a symbiotic relationship with brain development, brain health, and brain functioning [1]. With the growth of neuroscience discovery, policymakers and practitioner workforces have adopted ‘neuro-informed practice and policy’ in their decisions that target the delivery of human, social, and economic services that support human functioning and wellbeing. Yet, the plurality of concept definitions can lead to inaccurate or inconsistent operationalization and translation of findings from neuroscience into research, practice, and policy across fields and workforces. Precision in understanding the concept of NPP is necessary to ensure that any policy and practice actions are based on strong evidence and achieve their intended aims.

Our unifying definition of NPP and knowledge bases traverse implications for health and education outcomes, economic development, innovation, and justice, and highlight the key factors required for healthy societies beyond the neighborhood. Our definition and identified key knowledge bases reflect the growth in the literature focused on child development in the last two decades. This part of the literature encourages (1) deeper thinking beyond the brain in isolation to the interactive development of multiple biological systems and (2) moving beyond a focus on strengthening caregiver-child relationships to consider the broader social context of communities, government, business, and philanthropy working together to ensure supportive environments for ALL families raising children.

## 5. Limitations and Future Directions

Our review does not specifically address the outcomes of applying NPP, whether within a specific field or across broader systems, nor the quality of evidence to support these outcomes. Indeed, most examples of the implication of applying NPP identified within our scoping review were focused on narrow conceptualizations (e.g., brain training programs [52], clinical interventions [44]) or were largely speculative, without providing evidence of outcomes of these approaches. The current review also did not include papers published in languages other than English, thus potentially limiting understanding of the application of NPP across different cultural and language contexts. Finally, trauma-informed as defined by the Substance Abuse and Mental Health Administration (SAMSHA) [24,25] was used as a comparator for the development of a conceptual definition of NPP. It should be acknowledged, however, that the definition of trauma-informed is not itself without controversy [96], highlighting the challenges of the application of complex concepts into policy and practice actions.

The conceptual definition and 12 knowledge bases emerging from this scoping review provide a framework for future application and evaluation of NPP across fields and systems. Potential applications include but are not limited to informing the development of pre- and post-service training (e.g., curricular reviews, tertiary curricular development, and professional development), reviewing and informing policy and practice actions, and increasing public and community knowledge and awareness. In undertaking such implementation and evaluation, three key considerations are raised. First, NPP must incorporate a strong and rigorous neuroscience underpinning. NPP that is underpinned by outdated research evidence or applies over-simplistic interpretations and translations of evidence runs the risk of perpetuating a range of neuro-myths with the potential to do harm [97]. Consideration of the quality, applicability, and efficacy of evidence is necessary to ensure that NPP achieves its intended aims of creating and promoting optimal conditions for brain health and related positive physical, social, and community outcomes. The challenge of distilling, without oversimplifying, complex scientific concepts is not unique to NPP but reflects a larger challenge regarding the application of science into policy and practice action, e.g., [98]. Second, NPP must take into consideration the contexts within which individuals are situated. Conceptualizations of NPP that are focused on the brain in isolation, or the individual in isolation, disregard evidence of the complex and profound influence of social and physical environments and systems on brain development [99,100]. Finally, NPP should not be viewed as field-specific but as a unifying frame with the potential for application across systems.

## 6. Conclusions

Understanding the brain in context enables meaningful connections between scientific evidence and actions in policy and practice. Actions that are underpinned by a shared purpose, shared knowledge, and shared language targeting optimal child development have the potential to foster a society that not only nurtures future generations but also invests in its own long-term wellbeing and prosperity [64,101]. NPP, as defined in this paper, provides a new opportunity to consider the complex systems in which children and families live and the interplay between these systems and human functioning. That is, the connection from cells to society.

## Figures and Tables

**Figure 1 brainsci-14-01243-f001:**
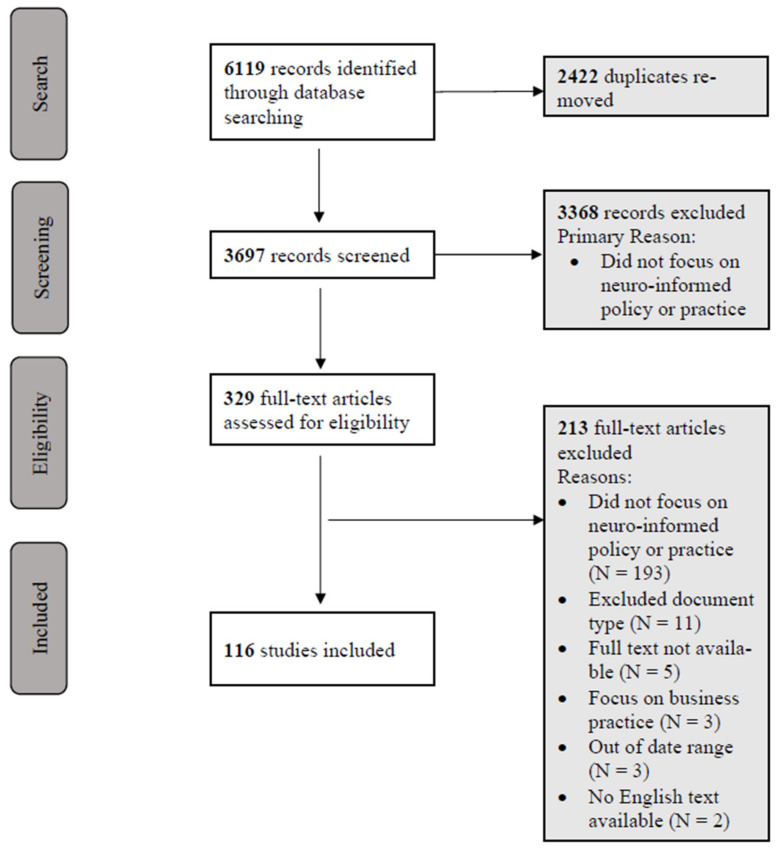
PRISMA flow diagram showing included studies at each stage of the inclusion/exclusion process.

**Figure 2 brainsci-14-01243-f002:**
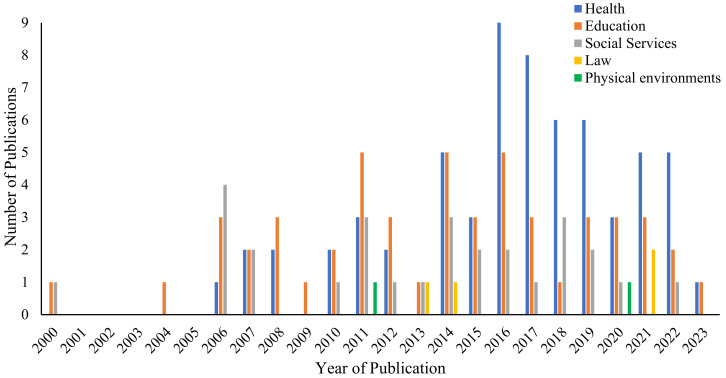
Number of publications referencing NPP, by year. *Note.* The search was conducted in March 2023.

**Figure 3 brainsci-14-01243-f003:**
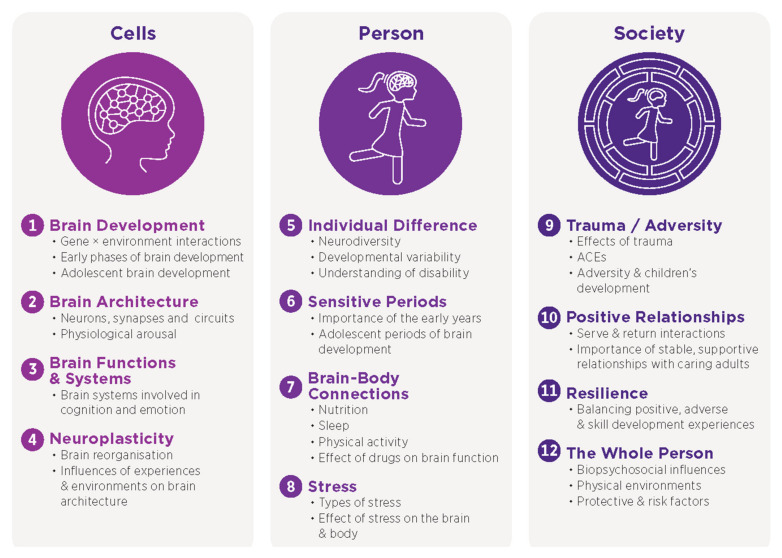
Key NPP knowledge bases and related examples. *Note*. The examples provided are indicative of content identified within each knowledge base across publications and are not intended to be exhaustive. Further examples and references to relevant publications for each knowledge base are provided in Appendix A.

**Table 1 brainsci-14-01243-t001:** Concepts and related definitions used for attribution comparison.

Concept Term	Definition(s) Used
Trauma-informed	“… a basic realization about trauma and understanding how trauma can affect families, groups, organizations, and communities as well as individuals” [24] (p. 9). A trauma-informed approach incorporates three key elements: “(1) realizing the prevalence of trauma; (2) recognizing how trauma affects all individuals involved with the program, organization, or system, including its own workforce; and (3) responding by putting this knowledge into practice” [25] (p. 4).
Evidence-Based	“Based on, concerning, or derived from evidence; empirical; (chiefly) spec. [specifically] designating an approach to medicine, health and social care, education, etc., which emphasizes the practical application of the findings of the best available current research in the field; of or relating to such an approach” [26] (evidence based).
Neuroscience	“The scientific study of the nervous system, including neuroanatomy, neurobiology, neurochemistry, neurophysiology, and neuropharmacology, and its applications in psychology, psychiatry, and neurology.” [27] (neuroscience).

**Table 2 brainsci-14-01243-t002:** Necessity and/or sufficiency of key attributes across NPP and comparative terms.

Key Attributes	Neuro-Informed	Trauma-Informed	Evidence-Informed	Neuro-Science	Conclusions
Is aligned with current research evidence	Present	Present	Present	Present	Necessary but not sufficient
2. Translates neuroscience	Present	Present	Absent	Present	Necessary but not sufficient
3. Guides (positive) actions	Present	Present	Present	Absent	Necessary but not sufficient
4. Describes brain-body-environment interaction	Present	Absent	Absent	Absent	Sufficient but not necessary
5. Explains (processes underpinning) human behavior	Present	Present	Absent	Present	Necessary but not sufficient
6. Applies knowledge	Present	Present	Present	Present	Necessary but not sufficient
7. Is development focused	Present	Present	Absent	Absent	Necessary but not sufficient
8. Connects science with practice/policy	Present	Present	Present	Absent	Necessary but not sufficient
9. Contains neuroscience messages and metaphors	Present	Absent	Absent	Absent	Sufficient but not necessary
10. Is brain health/brain development focused	Present	Absent	Absent	Present	Necessary but not sufficient
11. Creates optimal conditions	Present	Present	Present	Absent	Necessary but not sufficient
1 + 2 + 3 + 5 + 6 + 7 + 8 + 10 + 11	Present	Absent	Absent	Absent	Necessary and jointly sufficient

## Data Availability

A list of included papers from the scoping review can be found in Appendix A Appendix A.

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
