# Peer review of "The Brain in Context: A Scoping Review and Concept Definition of Neuro-Informed Policy and Practice"

_brainsci, 2024, doi:10.3390/brainsci14121243_

Round 1

Reviewer 1 Report

Comments and Suggestions for Authors

This paper addresses an important issue regarding the current absence of a clear definition for neuro-informed practice, meaning that this has been used to describe a variety of interventions and approaches. Establishing a definition is vital to ensure robust future scientific exploration and evaluation in this area; the paper makes a strong and coherent argument for the importance of concept definition and uses a robust methodology to achieve this.

The methodology appears to be robust, for the following reasons:

·         PRISMA guidelines followed

·         Prospective registration of the protocol

·         Range of search terms is broad and includes grey literature in addition to published work.

·         Utilised variety of databases and included different study designs.

·         Full texts screened by two independent authors. For transparency, I would suggest that the authors include the initials of the individuals who performed the screening phase.

Areas to be addressed by the authors:

It is unclear to me why the authors chose to identify five super-ordinate target areas for the application of NPP and the benefits of this section. Whilst they acknowledge some of these areas are not mutually exclusive, I am concerned that this could further reinforce ‘siloed’ thinking between e.g. health, education and social services which is in sharp contrast to the use of neuro-informed theory and application which emphasised the importance of interconnected thinking and services to really see an individual within the wider context of their family, environment and community. (Indeed, as stated by the authors themselves at the end of ‘Limitations and Future Directions’.)

The concept definition for Trauma-informed (Table 1) which is then used to map the key attributes of NPP in Table 2 is not itself without controversy. Indeed, there are multiple ways this term is used. The SAMHSA references are both missing from the reference list (which must be addressed). Whilst I acknowledge it is necessary to select a definition for trauma-informed (else the argument/process becomes circular) I think it is important for this to be acknowledged and explored within the paper.

Minor points:

·         The study is restricted to English-language publications, which could introduce bias.

·         The authors did not appear to assess the quality of the included studies, which I understand might be outside the scope of the paper. However, it would be helpful to consider how the quality of the studies might have influenced the findings.

·         Regarding the PRISMA flow diagram, I noticed a minor discrepancy: the full-text articles excluded add up to 217, not 213.

I am not familiar with content analysis methodology, so I am unable to comment on that aspect.

Author Response

Reviewer 1 – Comments

Comment 1

This paper addresses an important issue regarding the current absence of a clear definition for neuro-informed practice, meaning that this has been used to describe a variety of interventions and approaches. Establishing a definition is vital to ensure robust future scientific exploration and evaluation in this area; the paper makes a strong and coherent argument for the importance of concept definition and uses a robust methodology to achieve this.

The methodology appears to be robust, for the following reasons:

  • PRISMA guidelines followed
  • Prospective registration of the protocol
  • Range of search terms is broad and includes grey literature in addition to published work.
  • Utilised variety of databases and included different study designs.

Full texts screened by two independent authors. For transparency, I would suggest that the authors include the initials of the individuals who performed the screening phase.

Response 1

Thank you.

The initials of the individuals who performed the screening phase have now been included as suggested – on page 3.

Comment 2

Areas to be addressed by the authors:

It is unclear to me why the authors chose to identify five super-ordinate target areas for the application of NPP and the benefits of this section. Whilst they acknowledge some of these areas are not mutually exclusive, I am concerned that this could further reinforce ‘siloed’ thinking between e.g. health, education and social services which is in sharp contrast to the use of neuro-informed theory and application which emphasised the importance of interconnected thinking and services to really see an individual within the wider context of their family, environment and community. (Indeed, as stated by the authors themselves at the end of ‘Limitations and Future Directions’.)

Response 2

We appreciate the reviewers point here. The decision to use superordinate categories reflected the need to provide a framework for understanding the current application of NPP which is emerging across multiple fields and disciplines.

In response we have now clarified the purpose of this classification within the methods – on page 3.

Comment 3

The concept definition for Trauma-informed (Table 1) which is then used to map the key attributes of NPP in Table 2 is not itself without controversy. Indeed, there are multiple ways this term is used. The SAMHSA references are both missing from the reference list (which must be addressed). Whilst I acknowledge it is necessary to select a definition for trauma-informed (else the argument/process becomes circular) I think it is important for this to be acknowledged and explored within the paper.

Response 3

We thank the reviewer for raising this important point and have now included commentary regarding trauma-informed within the discussion – on page 11.

These are included as reference number 24 and 25.

Comment 4

The study is restricted to English-language publications, which could introduce bias.

Response 4

This has now been added as a limitation within the discussion section of the paper – on page 11.

Comment 5

The authors did not appear to assess the quality of the included studies, which I understand might be outside the scope of the paper. However, it would be helpful to consider how the quality of the studies might have influenced the findings.

Response 5

This has now been added as a limitation within the discussion section of the paper – on page 11.

Comment 6

Regarding the PRISMA flow diagram, I noticed a minor discrepancy: the full-text articles excluded add up to 217, not 213.

Response 6

These were reported correctly in the figure but incorrectly in text. We have amended the numbers stated in text and they are now accurate - end of page 4.

Comment 7

I am not familiar with content analysis methodology, so I am unable to comment on that aspect.

Reviewer 2 Report

Comments and Suggestions for Authors

I accepted review of this article because, from the abstract, it looked as though it would make a useful addition to the reading for one of the modules I teach on applications of neuroscience and it did not disappoint. Although ultimately a conceptual/definitions paper, the approach taken is rigorous and I think it is informative.

There are couple of minor points it would be great to see discussed briefly in the discussion section is possible:

- Two critiques are noted for NPP in 3.3. One or both of these have likely given rise to neuromyths which are not mentioned in the paper so acknowledging the literature in this space would be good. Additionally I wonder how unique these critiques are to neuroscience. Is this more about what happens when you try to apply any natural science to applied fields. Some discussion of this could be helpful.

- Three definitions are given in Table 2 (trauma, evidence and neuroscience) and I wondered if there was any relationship between these and the five fields identified. For example, were papers on education more likely to take an evidence informed approach.

- Related to this, Figure 3 shows the different types of knowledge that can be used in NPP and I wondered again what the relationship is between the conceptual/definition type, knowledge base and field of application. I know it is not likely to be clear cut but, for example, if most of the education work is adopting person level evidence then that might be noteworthy.

Author Response

Reviewer 2 – Comments

Comment 1

I accepted review of this article because, from the abstract, it looked as though it would make a useful addition to the reading for one of the modules I teach on applications of neuroscience and it did not disappoint. Although ultimately a conceptual/definitions paper, the approach taken is rigorous and I think it is informative.

Response 1

Thank you.

Comment 2

There are couple of minor points it would be great to see discussed briefly in the discussion section is possible:

Two critiques are noted for NPP in 3.3. One or both of these have likely given rise to neuromyths which are not mentioned in the paper so acknowledging the literature in this space would be good. Additionally, I wonder how unique these critiques are to neuroscience. Is this more about what happens when you try to apply any natural science to applied fields. Some discussion of this could be helpful.

Response 2

We have now indicated and included specific reference to neuro-myths within the critiques section – on page 7, and further discussion related to the application of science to policy and practice within the discussion section – on page 11.

Comment 3

Three definitions are given in Table 2 (trauma, evidence and neuroscience) and I wondered if there was any relationship between these and the five fields identified. For example, were papers on education more likely to take an evidence informed approach.

Response 3

These definitions were selected as comparators to support development of the NPP concept definition (as per the method described by Podsakoff, et al., 2016). The papers included were only those identified as using and/or defining NPP. Thus, the use of these other terms (trauma informed, evidence, neuroscience [in isolation) within the included papers was not examined.

Comment 4

Related to this, Figure 3 shows the different types of knowledge that can be used in NPP and I wondered again what the relationship is between the conceptual/definition type, knowledge base and field of application. I know it is not likely to be clear cut but, for example, if most of the education work is adopting person level evidence then that might be noteworthy.

Response 4

The reviewer raises an interesting point. The aim of the current paper was to focus on the range of knowledge bases being applied across included papers and we intentionally chose not to report by group as we felt this would potentially lead to misunderstanding that some knowledge bases were only relevant to specific sectors/fields. The distribution of knowledge bases by focus area is difficult to disentangle as the papers (irrespective of focus area) almost always include multiple knowledge bases thus making it difficult to separate in any meaningful way across areas. We have therefore not included at this time.
